# The Role of Extracellular Matrix Proteins in Breast Cancer

**DOI:** 10.3390/jcm11051250

**Published:** 2022-02-25

**Authors:** Arkadiusz Lepucki, Kinga Orlińska, Aleksandra Mielczarek-Palacz, Jacek Kabut, Pawel Olczyk, Katarzyna Komosińska-Vassev

**Affiliations:** 1Department of Community Pharmacy, Faculty of Pharmaceutical Sciences in Sosnowiec, Medical University of Silesia in Katowice, 41-200 Sosnowiec, Poland; alepucki@interia.pl (A.L.); kinga.orlinska@sum.edu.pl (K.O.); 2Department of Immunology and Serology, Faculty of Pharmaceutical Sciences in Sosnowiec, Medical University of Silesia, 41-200 Sosnowiec, Poland; apalacz@sum.edu.pl (A.M.-P.); jacekkabut@poczta.onet.pl (J.K.); 3Department of Clinical Chemistry and Laboratory Diagnostics, Faculty of Pharmaceutical Sciences in Sosnowiec, Medical University of Silesia, 41-200 Sosnowiec, Poland; kvassev@sum.edu.pl

**Keywords:** breast cancer, extracellular matrix, proteins, tumorigenesis, tumor microenvironment

## Abstract

The extracellular matrix is a structure composed of many molecules, including fibrillar (types I, II, III, V, XI, XXIV, XXVII) and non-fibrillar collagens (mainly basement membrane collagens: types IV, VIII, X), non-collagenous glycoproteins (elastin, laminin, fibronectin, thrombospondin, tenascin, osteopontin, osteonectin, entactin, periostin) embedded in a gel of negatively charged water-retaining glycosaminoglycans (GAGs) such as non-sulfated hyaluronic acid (HA) and sulfated GAGs which are linked to a core protein to form proteoglycans (PGs). This highly dynamic molecular network provides critical biochemical and biomechanical cues that mediate the cell–cell and cell–matrix interactions, influence cell growth, migration and differentiation and serve as a reservoir of cytokines and growth factors’ action. The breakdown of normal ECM and its replacement with tumor ECM modulate the tumor microenvironment (TME) composition and is an essential part of tumorigenesis and metastasis, acting as key driver for malignant progression. Abnormal ECM also deregulate behavior of stromal cells as well as facilitating tumor-associated angiogenesis and inflammation. Thus, the tumor matrix modulates each of the classically defined hallmarks of cancer promoting the growth, survival and invasion of the cancer. Moreover, various ECM-derived components modulate the immune response affecting T cells, tumor-associated macrophages (TAM), dendritic cells and cancer-associated fibroblasts (CAF). This review article considers the role that extracellular matrix play in breast cancer. Determining the detailed connections between the ECM and cellular processes has helped to identify novel disease markers and therapeutic targets.

## 1. Introduction

Female breast cancer is the leading cause of global cancer incidence, with an estimated 2.3 million women diagnosed with breast cancer and 685,000 deaths globally. In the end of 2020, there were 7.8 million women who had been diagnosed with breast cancer in the last 5 years, making it the most common cancer in the world. Breast cancer mainly involves the inner layer of the milk glands or lobules and ducts (small tubes that carry milk) [1,2,3]. Adipose tissue is present in the mammary gland (both female and male). The amount of fat determines the size of the breast [2,4]. There are some differences in the microarchitecture of the female and male mammary gland. In men, less glandular tissue is found in the glands than in women [2,5]. The female breast generally contains 12–20 lobes. These, in turn, are divided into smaller elements called lobules [2,6]. These lobes and lobules are connected via milk ducts. The adipose tissue of the breast is supplied by a network of lymph vessels, lymph nodes, blood vessels and nerves. The breast is also composed of fibrous connective tissue and ligaments which ensure its proper shape [2,7]. Hormonal changes that occur in the female body during the menstrual cycle, pregnancy and puerperium have a significant impact on the structure and function of the mammary gland. One of the hormonal effects of nipple stimulation is secretion of prolactin, which is produced by acidophilic cells (called lactotrophs or mammotrophs) in the anterior lobe of the pituitary gland [2,8]. The epidermis of the areola and nipple is characterized by moderate wrinkling as well as marked pigmentation. The nipple skin contains somewhat small hair and several apocrine and sebaceous sweat glands [2]. Milk sinuses are formed at the base of the nipple by milk ducts (usually 15–25 in number). These ducts transport milk towards the nipples [2,9]. Slightly under the surface of nipple, these sinuses end in coneshaped ampullae. The spherical areola is present around the nipple and is 15–60 mm in diameter [2]. Deep in the areola and nipple, several smooth muscle fibers are set radially and circularly in the dense connective tissue and longitudinally alongside the lactiferous ducts that lengthen up into the nipple. The muscle fibers mentioned above are involved in nipple erection, contraction of areola as well as emptying of milk sinuses [2].

## 2. ECM

The extracellular matrix (ECM), which is a perfectly organized and efficiently managed structure, is formed from a great variety of macromolecules, forming a multitude of combinations, depending on the tissue in which this structure occurs. It can be regarded as a physical scaffold for cellular components, although the range of functions it performs is much broader, and many of them are not, as yet, believed to be known and described. The proper combination of its components not only ensures appropriate stability and durability of the ECM, but most importantly determines the mechanical properties of a given tissue and serves as a bioreservoir for molecules such as growth factors. The role of ECM in many processes essential for cell homeostasis has been documented, including: adhesion, apoptosis, proliferation, differentiation, and migration. Genetically determined disorders of ECM structure or function have been shown to disrupt tissue and systemic homeostasis followed by various diseases. The composition of the ECM in a given tissue is determined during its development by a biochemical dialogue between the cells and the environment. This composition is an expression of its adaptation to the function performed in the body [10,11,12,13,14,15]. This section focuses on describing the general properties of ECM proteins, while their role in breast cancer will be discussed in detail in Section 3.4.

### 2.1. Collagen

Collagen accounts for nearly 30% of the total protein mass found in animals [12,15,16,17,18]. In humans, it makes up about 75% of the dry weight of the skin and is the most common component of the ECM [8]. Its essential functions include: maintaining the structural integrity of tissues, participating in wound healing, regulating cell adhesion, enhancing chemotaxis, promoting migration, providing tensile strength to tissues, and in addition, overseeing the proper course of their development and differentiation [12,16].

Collagen is formed from three left-handed polypeptide α chains, organized into a triple helical structure that is right-handed [10,12,17,18,19]. The described filamentous protein can be either a homotrimer or a heterotrimer [18,19,20,21]. In vertebrate animal organisms, 46 chains have been identified that can organize into 28 different collagen types [10,17,18,19,20,21]. These include: fiber-forming collagens (e.g., types I, II, and III), network-forming collagens (e.g., type IV basement membrane collagen), collagens associated with fibrils with breaks in their triple helixes (e.g., types IX and XII), and others (e.g., type VI) [10,17,18,21]. The tight packing of the trihelical structure is possible due to the presence of the characteristic Gly-X-Y motif (Gly—glycine), an amino acid sequence that repeats multiple times in the helix-forming polypeptide chains. Glycine is crucial for the stability of this structure because it is the only one of all the amino acids that is small enough to fit into the central part of the core of the helix described above, into which every third of the amino acid residues building each of the polypeptide α chains enters. The role of electrostatic interactions, inter-chain hydrogen bonds, and high proline and hydroxyproline content in maintaining the stability of the collagen triple helix is also emphasized [10,17,18,19,20,21,22].

The process of collagen biosynthesis has been most thoroughly studied and described for fiber-forming collagens [21]. They are synthesized in the form of precollagens, which contains a signal sequence that is cleaved after synthesis in the endoplasmic reticulum. After that, the resulting procollagen molecules contain: an amino-terminal propeptide followed by a short, non-helical N-telopeptide, a central triple helix, a C-telopeptide and a carboxyl propeptide. Individual procollagen molecules can be post-translationally modified in various ways. This is done by: hydroxylation of some proline and lysine residues, glycosylation of some hydroxylysine residues as well as sulphation of tyrosine residues [21,23]. The adoption of the final, stable conformation by collagen depends on the proper attachment to procollagen of a specific molecular chaperone—heat shock protein 47 (Hsp47)—in the endoplasmic reticulum [21,24]. It has been established that for procollagen to be adequately stable at human body temperature, more than 20 Hsp47 molecules per triple helix must be attached [21,25]. Hsp47 is believed to protect procollagen from random unfolding and uncontrolled aggregation. It is also responsible for proteostasis regulation (folding, quality control, secretion) [18,26]. Intracellular Secreted Protein Acidic and Rich in Cysteine (SPARC) also pretends to be a chaperone of procollagen. It has been observed that the lack of this protein or its dysfunction result in defective deposition of collagen in tissues. Its ability to bind to the triple-helical structure of procollagen has also been demonstrated [21,27]. Both propeptides that make up procollagen require removal in a process called maturation [21,28]. The N-propeptide is cleaved by procollagen N-proteinases belonging to the A Disintegrin and Metalloproteinase with the Thrombospondin Motifs (ADAMTS) family, except the N-propeptide of the proα1(V) chain that is cleaved by the procollagen C-proteinase also termed Bone Morphogenetic Protein-1 (BMP-1) [21,29]. BMP-1 cleaves the carboxy-terminal propeptide of procollagens, except the carboxy-terminal propeptide of the proα1(V) chain, that is processed by furin. The telopeptides contain the sites where cross-linking occurs. The mentioned linkage formation is initiated by the oxidative deamination of lysyl and hydroxylysyl residues catalyzed by the enzymes of the lysyl oxidase (LOX) family [21,30].

It is widely believed that the unique mechanical properties of fiber-forming collagens are determined by covalent crosslinks. A special role in this regard is attributed to reducible and mature crosslinks produced via the LOX pathway. Cross-linking is viewed as tissue-specific rather than collagen-specific. Maturation of cross-links has been shown to provide additional resistance to shear stress [21].

Collagen degradation occurs essentially through the catalytic activity of matrix metalloproteinases (MMPs). These are zinc-dependent endopeptidases. They belong to the metzincin superfamily. They may take part both in physiological processes (e.g., development of tissues and their repair after damage) and in pathological processes (e.g., metastasis of neoplastic cells). Fiber-forming collagens (i.e., types I, II, and III) are substrates for: MMP-1 (also known as interstitial collagenase), MMP-8 (whose other name is neutrophil collagenase), MMP-13 (also referred to as collagenase 3), and MMP-14 (membrane-anchored). As far as collagen type I is concerned, it should be mentioned that it can also be degraded by MMP-2. The preferential substrates for particular zinc-dependent endopeptidases are collagen types I and III (for MMP-1 and MMP-8), and collagen type II for MMP-13 [21,31]. MMP-2 and MMP-9 are involved in the degradation of collagen type IV as well as denatured collagen (other names of these enzymes are 72 kDa-gelatinase and 92-kDa-gelatinase, respectively). It is noteworthy that the [α1(I)]3 homotrimer of collagen type I, in contrast to the [α1(I)]2 α2(I) heterotrimer of the same collagen, is not a substrate for mammalian MMPs, which is explained by the resistance of the homotrimer to local triple helix unwinding by MMP-1 due to the higher triple helix stability near the MMP cleavage site [21,32]. MMPs also play a key role in releasing of bioactive fragments or matricryptins (e.g., tumstatin or endostatin—two inhibitors of angiogenesis) from full-length collagens. Another group of enzymes releases the ectodomain of membrane collagens as soluble forms. These enzymes are collectively called sheddases [21,23,33,34,35].

It has been established that the effect of collagens on cell–matrix interactions is mediated by receptors. Several families of them have been identified [21,36,37,38,39]. These receptors are ligands for integrins, cell-adhesion receptors that lack intrinsic kinase activities. The ability to bind collagen is demonstrated by integrins that have a β1 subunit connected with one of four subunits (i.e., α1, α2, α10 or α11) characterized by the presence of a domain known as αA. The discussed linkage is made by GFOGERlike (GFOGER—glycine–phenylalanine–hydroxyproline–glycine–glutamic acid-arginine) sequences [21,36,38]. Recognition sequences other than the mentioned one (e.g., KGD—lysine–glycine–aspartic acid) have also been identified in some collagens. Integrins αvβ1 and α5β1 have the ability to bind collagen type XVII, which exposes the KGD sequence in its ectodomain [21,36]. Not only can collagens be ligands for integrins, but also their proteolysis products, as confirmed for the following integrins: α3β1, α5β1, αvβ3 and αvβ5 [21,33]. Dimeric discoidin receptors (DDR1 and DDR2) that possess tyrosine kinase activities can also bind collagens. This has been observed for collagen types I, II and III [21,39]. DDR1 is widely expressed in epithelial cells, while DDR2 is mainly found in mesenchymal cells. The major DDR2-binding site in collagens I-III is a GVMGFO (glycine–valine–methionine–glycine–phenylalanine–hydroxyproline) motif [21,37]. It is assumed that collagen binding triggers structural reorganization of DDR2 surface loops, which leads to an activation of discoidin domains, and it is worth highlighting that mentioned domains can independently bind to collagen or simultaneous binding of two domains to the protein triple helix can occur [21,40]. Soluble extracellular domains (DDR1 and DDR2) also promote collagen deposition in the ECM by blocking fibrillogenesis, and in addition DDR2 determines the mechanical properties of collagen type I fibrils [21,40,41,42]. Collagen binding on platelets is mediated by glycoprotein VI (GPVI). It is a member of the paired immunoglobulin-like receptor [21,36]. Collagen can also bind to the inhibitory leukocyte-associated immunoglobulin-like receptor-1 (LAIR-1) [21,43]. Both GPVI and LAIR-1 are capable of recognizing the GPO (glycine–proline–hydroxyproline) motif in collagens. Membrane collagens (XIII, XVII, XXIII) and fibrillar collagens (I, II, III) act as ligands for LAIR-1, and it is worth noting that type I and III collagens are so-called functional ligands and block the activation of immune cells in vitro. LAIR-1 has multiple binding sites on collagen types II and III. GPVI has lower affinity for collagens type I and III than LAIR-1 [12,43,44,45]. Three LAIR-1 amino acids central to collagen binding are conserved in GPVI [12,46]. Fibril-forming collagens and collagen type IV are also ligands of Endo180 (urokinase-type plasminogen activator associated protein), a member of the macrophage mannose-receptor family that mediates collagen internalization [21,37,39].

### 2.2. Elastin

In mammals, elastin is encoded by a single gene and secreted as a 60–70 kDa tropoelastin monomer. Its primary role is to provide elasticity and resilience to tissues that are repeatedly stretched. It is found in tendons, stretchers, ligaments, walls of major blood vessels (e.g., abdominal aorta) and lung tissue [10,13,17,47]. Importantly, elastin stretch is crucially limited by tight association with collagen fibrils [17,48]. Fibulins enable tropoelastin to associate with microfibrils. In this way, elastic fibers are formed. A characteristic feature of all tropoelastin structures is the presence of hydrophobic sequences alternating with lysine-containing cross-linking motifs. Fibrillins and microfibril-associated glycoprotein-1 play important roles in the nucleation and assembly of elastin. An essential role in providing and maintaining the characteristic mechanical properties of elastin is attributed to the extensive cross-linking of tropoelastin, which is catalyzed by LOX [10,13,47,49]. This enzyme oxidizes selective lysine moieties in peptide bonds to allysine. There are two bifunctional cross-links in elastin: dehydrolysinonorleucine and allysine aldol. The former one is formed through the condensation of one residue of allysine and one of lysine. The latter one is formed through the association of two allysine residues. These two cross-links can further condense with each other or with other intermediates to form desmosine or isodesmosine, being the major cross-links of the mature elastic protein. It has been documented that tropoelastin manifests the ability to form globular structures (aggregates) on the cell surface in a process called microassembly. Cross-linking implies the loss of positive charges on the molecule, which promotes the release of tropoelastin from cells, as well as facilitates global fusion in the presence of microfibrils (macroassembly). Fibulin-4 plays a key role in early stages of elastin assembly, whereas fibulin-5 acts to bridge elastin between the matrix and cells [10,13,47,49].

### 2.3. Laminin

Laminins form a family of heterotrimeric (one α chain, one β chain, and one γ chain) glycoproteins that includes nearly 20 members. Laminins are assembled into a cross-linked web. In basement membranes, this web is intertwined with a network that is composed of collagen fibrils. The mass of heterotrimers oscillates in the range of 400–800 kDa. It was found that in vertebrate animals there are five α chains and three β and γ chains each [10,50,51,52]. They are essential for normal organogenesis. They are also involved in early embryonic development [53,54]. For many of the known laminins, the ability to form networks spontaneously through appropriate connections has been documented. Such structures are able to interact with receptors located on cell surfaces [10,50,51,52,53,54,55,56,57,58].

### 2.4. Fibronectin

Fibronectin acts as a biological glue, participating in the management of the functions and structure of the interstitial ECM and being involved in facilitating target attachment and promoting cell migration [10,12]. The building units of monomers of this protein are subunits that contain three types of repeats (I, II and III). The average mass of such a monomer is 250 kDa. Fibronectin is secreted as sulfide-linked dimers. It has binding sites to other fibronectin dimers, cell surface receptors, heparin and collagen [10]. Cellular traction forces can stretch fibronectin several times over its resting length. This favors the exposure of cryptic integrin binding sites within the molecule, resulting in pleiotropic changes in cellular behavior. For this reason, fibronectin has been implicated as an extracellular mechanoregulator [12,59]. The fibronectin dimers can form multimers. Further fibronectin deposition is accompanied by structural changes (thickening and elongation) of fibrils. Fibronectin fibrils can be further processed into a deoxycholate-insoluble matrix [10,60]. Fibronectin plays a significant role in cell migration (both during embryonic development and during wound healing) [12,61,62,63].

### 2.5. Proteoglycans

Proteoglycans (PGs) are formed from a protein core and laterally attached glycosaminoglycan (GAG) chains. GAGs (hyaluronic acid, heparan sulfate/heparin, dermatan sulfate, chondroitin sulfate, keratan sulfate) are a heterogeneous group of anionic polysaccharides with characteristic disaccharide units (amino sugar, uronic acid or galactose) that are repeated many times in their structure. All except hyaluronic acid are sulfated [10,12,15,64]. It is believed that the negatively charged structure of GAG chains is crucial for the ability of PGs to sequester divalent cations and water. It is on this sequestration that a wide range of functions performed by PGs in tissues (e.g., lubrication functions and conferring space-filling) are commonly believed to depend. Based on the structure of core proteins, localization and composition of GAGs, PGs have been divided into three most important groups: cell-surface PGs, modular PGs and small leucine-rich PGs [10,12,15]. The functional diversity of PGs is based on their molecular diversity. It is worth emphasizing that small leucine-rich PGs participate in multiple signaling pathways (including binding to and activation of low-density lipoprotein-receptor-related protein 1, insulin-like growth factor 1 receptor and epidermal growth factor receptor). They are also involved in the inflammatory response reaction due to their ability to bind and activate transforming growth factor β [10,12,15,65,66,67]. Cell-surface PGs (glypicans and syndecans) improve the course of ligand–receptor interactions, so they are attributed to the role of co-receptors [12,68]. Basement membrane modular PGs (agrin and perlecan) can act as both pro- and antiangiogenic molecules [10,12,15]. Modular PGs co-supervise cell proliferation, migration and adhesion [12,15]. The importance of PGs in collagen fibril assembly is also emphasized [10].

### 2.6. Thrombospondin

In vertebrate animals, thrombospondins are encoded by a THBS gene family consisting of five members. The function of thrombospondin 1 (THBS1) is best understood (under both physiological and pathological conditions). The precursor of this protein is composed of 1170 amino acids, whereas the mature protein, which is devoid of the N-terminal signal peptide compared to the aforementioned precursor, undergoes homotrimerization after secretion. THBS1 comprises roughly twelve asparagine-linked mono-, bi- tri-, and tetraantennary complex oligosaccharides and variable numbers of C-mannosylated tryptophan residues in the type 1 repeats. The protein in question was also found to be O-fucosylated [69,70,71,72,73,74]. During prenatal development, THBS1 is expressed in many tissues, whereas in adults who do not suffer from cancer, the expression of this protein is low. It has been pointed out that there is a positive correlation between age and plasma THBS1 levels. Moreover, the association of elevated plasma concentrations of this protein with diseases typically associated with old age, such as cardiovascular disease and type 2 diabetes, is emphasized. It is worth noting that THBS1 is the most abundant thrombocyte alpha granule protein. Its plasma concentrations in healthy people are low. There is evidence that stimuli such as: ischemia, tissue remodeling, injury or reperfusion are able to induce THBS1 expression in many locations in the body. Its high plasma concentrations have been reported in patients with rheumatoid synovitis, atherosclerosis and glomerulonephritis. Moreover, high plasma concentrations of lipids and glucose have been found to trigger THBS1 expression. It should be emphasized that increased expression of this protein occurs in the stroma of many cancers. The presence of THBS1 in ECM is transient owing to the fact that endotheliocytes and fibroblasts have the ability to efficiently internalize and degrade this protein. In the subendothelial matrix of some blood vessels as well as at the dermal–epidermal boundary in the skin, THBS1 expression is constitutive [69,75]. From a practical standpoint, the most important molecules that can bind THBS1 are: cathepsin G, fibronectin, some MMPs, fibrinogen, some collagens, active and latent transforming growth factor β1, plasmin, neutrophil elastase, tissue factor pathway inhibitor and heparin [69,76]. In a cell-specific as well as context-dependent manner, THBS1 can stimulate or inhibit proliferation, adhesion, motility, and survival of the cells. This protein is currently recognized as a potent inhibitor of angiogenesis. Nevertheless, the N-terminal proteolytic and recombinant parts of THBS1 have been found to stimulate angiogenesis, with integrin β1 mediating this effect. It has been demonstrated that THBS1 can block the enzymatic activity of proteases such as: neutrophil elastase, cathepsin G, and plasmin. Latent transforming growth factor β1 is in turn stimulated by THBS1. It is also worth highlighting that THBS1 has the ability to block stem cell self-renewal [69,77,78].

### 2.7. Osteopontin

Osteopontin (OPN) is a glycosylated extracellular matrix phosphoprotein produced by cells such as: osteoblasts, osteoclasts, epitheliocytes, endotheliocytes and immune cells. Depending on the tissue in which it is found, this protein can exhibit both structural and functional heterogeneity. The mass of OPN oscillates between 41 and 75 kDa, which is mainly due to differences in its post-translational modifications [79,80,81,82,83,84,85,86]. Bone remodeling, vascularization, inflammation as well as immune-regulation are processes in which OPN plays an important role. At this point it is worth emphasizing that this protein is also relevant in the course of tumorigenesis [79,87,88,89,90,91]. Data collected so far indicate that OPN together with some integrins, when stimulated by vascular endothelial growth factor (VEGF), may enhance angiogenesis. It is assumed that OPN promotes proliferation and migration of endothelial cells [79,92,93]. Many of the functions that OPN performs in the body are linked to its interactions with CD44 and integrins. Due to these interactions, OPN has been classified, along with dentin matrix protein 1 (DMP1), dentin sialophosphoprotein (DSPP), bone sialoprotein (BSP) and matrix extracellular phosphoglycoprotein (MEPE), into a group of molecules referred to as small integrin-binding ligand N-linked glycoproteins (SIBLINGs). Arginine–glycine–aspartic acid (RGD) and serine–valine–valine–tyrosine–glutamate–leucine–arginine (SVVYGLR) are two sequences that are necessary for the integrin-binding ability manifested by OPN. The former sequence enables OPN to bind to the following integrins: αvβ1, αvβ3 and αvβ5, while the latter one conditions OPN to bind to the following integrins: α4β1, α4β7, and α9β1 [79,94,95,96,97]. The binding of OPN to CD44 and integrins triggers a downstream signaling cascade via the PI3K/AKT signaling pathway leading to cell proliferation and survival which is mediated by NF-κB. Furthermore, via the Ras/Raf/MEK/ERK signaling pathway, an OPN–integrin complex confers a metastatic phenotype on some cancer cell types which appears to be dependent on the induction of activator protein 1-dependent gene expression [79,98,99,100,101,102,103].

### 2.8. Osteonectin

Osteonectin (ON) is a 32 kDa calcium-binding matricellular protein. It has been proven that this protein can be expressed in both mineralized and non-mineralized tissues, although initially many researchers believed that ON expression did not occur in the latter. As a rule, osteopontin expression accompanies the expression of fiber-forming collagens (e.g., collagen type I). It is widely believed that the role of ON in osteoid consists of the release of calcium cations on the one hand and the binding of both collagen and hydroxyapatite on the other. This constellation of ON properties is an assumption for its recognition as a promoter of bone mineralization. It is assumed that spatial separation of two domains of this protein, i.e., the hydroxyapatite-binding domain and collagen-binding domain, is essential for ON to properly perform this function [104,105,106,107]. ON is encoded by a single gene. This protein has four domains: an N-terminal low-affinity, high capacity, calcium-binding domain that contains the mineral binding region, a cysteine-rich domain, a hydrophilic region, and an extracellular Ca^2+^ (EC) domain with an E-F hand motif at the C-terminus that encompasses the collagen binding domain. ON undergoes differential glycosylation, which depends mainly on its tissue-specific expression. It is worth highlighting that the glycosylated form of this protein is expressed in bone and has a higher affinity for collagen than the form found in thrombocytes [104,108]. It has been discovered that ON can also be produced by fibroblasts and endotheliocytes. Moreover, this protein is found in platelet granules during injuries [104,109,110,111]. ON facilitates procollagen processing by limiting procollagen association with cell surface receptors, while noting that the ON-binding site on collagen overlaps with that of the collagen receptors called DDR1 and DDR2. It has been suggested that, by binding to collagen type I, ON may block signaling pathways mediated by DDR2. It is speculated that, in the absence of ON, streamlined interactions between soluble collagen and DDR2 entail increased turnover of collagen within the cell surface, resulting in changes in its deposition in the ECM. In addition, ON affects the content and diameter of collagen fibrils in mineralized tissues by regulating the activity of the enzyme called transglutaminase [21,37,39,40,41,42,104,112,113,114,115].

### 2.9. Periostin

Periostin (POSTN), which was originally isolated from a mouse osteoblast cell line as osteoblast-specific factor 2, belongs to the matricellular protein family. In humans, POSTN is encoded by the POSTN gene, whose expression can be increased by interleukins (4 and 13), and by transforming growth factor β. The components of the POSTN molecule are: a cysteine-rich domain within the N-terminal region, four fasciclin I domains, and an alternative splicing domain within the C-terminal region. It is worth adding that up to nine splice variants have been identified, but the full-length transcript encodes an approximately 90 kDa secreted protein that includes all exons. It is still unclear what functional significance these variants may have. It is conjectured that they might condition the differential expression of POSTN in different tissues and diseases [116,117,118,119,120,121]. POSTN binds both fibronectin and collagen type I and participates in collagen fibrillogenesis [116,122]. It has been proven that such known actions of POSTN as: promotion of adhesion, stimulation of proliferation, enhancement of angiogenesis, facilitation of metastasis or acceleration of cell migration are dependent on its binding to appropriate integrin receptors [116,117,118,123]. It has been reported that POSTN is engaged in epithelial–mesenchymal transition (EMT) and heart morphogenesis. The described protein is also relevant in such processes as Th2-dependent immune response and inflammation [124].

### 2.10. Tenascin C

Hexameric tenascin C (TNC) is a member of the tenascin gene family of proteins. It was originally attributed to function as a supervisor of cell adhesion. Nowadays it is known that the range of functions performed by this protein is much wider and includes, e.g., regulation of signaling between cells, modulation of expression of specific genes or participation in maintenance of proper biochemical conditions within the cellular microenvironment [125,126]. Due to differences in post-translational modifications, the described protein, which in vertebrates is characterized by a highly conserved amino acid sequence, is found in different molecular forms. Wide distribution of TNC occurs in embryonic tissues, while in postnatal development its expression is much lower and its synthesis is tightly regulated by many factors. However, there are some situations in which TNC synthesis is enhanced postnatally, e.g., during tissue healing after damage, during carcinogenesis (especially in the stroma of solid tumors), and during inflammation (especially chronic) [125,126,127,128,129]. The high prenatal expression of TNC is suspected to be related to the experimentally demonstrated ability of TNC to influence cell phenotype by interacting with appropriate receptors on the cell surface. Several glycosylation sites within the TNC molecule have been identified. It is suggested that TNC acquires protease resistance due to this modification. There are also reports that glycosylated TNC finds it more difficult to form hexamers. It has been also documented that glycosylation of TNC results in its ability to promote neuronal stem cell proliferation. Another post-translational modification that TNC may undergo is citrullination, which is attributed to the ability to increase the immunogenicity of C-terminal residues of this protein, which in turn leads to the formation of autoantibodies, as described in the case of rheumatoid arthritis [130,131,132]. TNC can be cleaved by gingipain cysteine proteases and MMPs. This degradation not only regulates TNC turnover in tissues. Indeed, it has been shown that the resulting molecules (soluble fragments) also have specific biological activities, usually different from the parent molecule. For instance, cleavage uncovers cryptic pro-apoptotic activity, hidden fibronectin-binding sites and concealed heparin–sulphate-binding sites that promote cell spreading [125,126].

### 2.11. Entactin

Entactin is a sulfated, multidomain glycoprotein that is found in many basement membranes. This protein is composed of 1217 amino acids that form two globular domains, linked by a rod-like structure whose essential fragments are four EGF- and one thyroglobulin-like cysteine-rich homology repeats. Entactin has the ability to bind to the following molecules: fibronectin, laminin, fibrinogen and collagen type IV. The described protein plays an important role in endowing the ECM with its proper characteristics. Entactin can promote both phagocytosis and chemotaxis. It is worth noting that these actions are dependent on interactions with integrin receptors. Entactin is also engaged in regulating wound healing and hemostasis by binding to fibrinogen. This binding is not dependent on metal cations. In addition, it has been shown that entactin is involved in controlling cell adhesion, which may be important for its role in tumorigenesis [133,134].

### 2.12. ECM in the Breast

In a resting adult mammary gland, the basement membrane encapsulates the gland and is the principal ECM that interacts with both the myoepithelium and the luminal epithelium [135]. Its essential components are: collagen type IV, PGs, laminins (111 and 332), entactin and epiligrin. ECM is responsible for maintaining the proper polarity by the epithelial cells, and it should be mentioned that this function is realized mainly due to the presence of the aforementioned laminin 111. The appropriate biochemical dialogue between ECM components and lactogenic hormones is required for full differentiation of mammary epithelial cells [135,136,137,138,139]. Signal transducer and activator of transcription 5 (STAT5) participates in this differentiation and in the process of milk secretion from the breast [135,140]. Inhibition of control of mammary epithelial differentiation in response to prolactin and impaired milk secretion occur when the epithelial cells are placed on the interstitial matrix that is rich in fibrillar collagen (type I and III), PGs, hyaluronan and various glycoproteins [135,141]. The presence of laminin 111 allows this control to be regained, but only if integrin β1 (receptor for laminin 111) is present and has no defects in structure or function. In its absence, STAT5 signaling is impaired and epithelial cells detach from the basement membrane. This integrin is also required for mammary ductal cells to proliferate. Extensive reorganization of the ECM in the mammary gland is observed during pregnancy and lactation. The post-lactational involution occurs with a significant increase in fibrillar collagen and fibrillin content. Increased proteolysis is also observed. Laminin, collagen type IV and entactin are degraded [135,142,143,144,145,146].

## 3. Breast Cancer

Breast cancer is one of the most common cancers affecting women worldwide, and its incidence continues to increase [147,148,149,150,151,152,153,154,155,156,157,158]. It is also one of the leading causes of cancer deaths among women [147]. As a metastatic cancer, it can exhibit the ability to spread to multiple organs (e.g., brain, lungs, kidneys, bones, and liver), which significantly worsens the prognosis [148,155]. In the absence of any metastases, the cure rate of patients sometimes reaches 90%, whereas, in the metastatic setting, the cure is not achievable for now. It is believed that in such cases the long-term survival depends mainly on the organs to which the metastases occur, as well as the extent and speed with which it occurs [157]. Early diagnosis facilitates treatment and is associated with a better prognosis [148]. Appropriate prophylaxis is also not without importance [154]. Many risk factors have been identified, including: age, gender, genetics, cigarette smoking, personal and family history, breast pathology (especially proliferative breast disease), reproductive factors, as well as dietary habits and estrogen metabolism disorders [147,148,149,150,151,152,153,154,155,156,157,158]. At this point, it is worth noting that mutations in two genes (BRCA1 and BRCA2) are the most significant causes of genetically determined breast cancer [149].

### 3.1. Molecular Subtypes of Breast Cancer

Based on the expression profile of specific genes, supported by immunohistochemical assays, the following molecular subtypes of breast cancer were identified: luminal A, luminal B, luminal HER2, enriched HER2 and triple-negative [2,3,4,135,137].

Luminal A subtype is the most common (accounting for nearly 50% of all newly diagnosed breast cancer cases) subtype and also the least aggressive [2]. It shares some features with luminal breast epithelial cells, namely, it manifests high expression of cytokeratins (7, 8, 18 and 19). Moreover, the expression of proliferation-stimulating genes is low in this subtype (low Ki-67 index), while the prognosis is very good. Lymph node involvement is rare, and the clinical course is relatively benign. It expresses estrogen receptor (ER) and progesterone receptor (PR), however the expression of human epidermal growth factor receptor 2 (HER2) is very low in this subtype. This constellation of properties makes luminal A subtype susceptible to hormonal therapy (using aromatase inhibitors or selective estrogen receptor modulators) [2,3,4,135,137].

Luminal subtype B accounts for 20–30% of invasive breast cancer cases and it is worth noting that most breast cancers genetically determined by BRCA2 gene mutation belong to this subtype [149,150]. This subtype is characterized by: high expression of cytokeratins, intermediate prognosis, higher risk of local recurrence after treatment (than in the case of subtype A). It is assumed that higher (than in subtype A) expression of Ki67, cyclin E1 and nuclease sensitive element binding protein 1, indicating increased proliferation, implies worse prognosis than in subtype A. It seems that increased signaling via pathways involving Src and PI3K kinases is also significant in this regard. This subtype is usually treated as the most aggressive form of hormone-dependent breast cancer. In addition to hormone therapy, this subtype usually requires additional treatment options: targeted therapy (if the cancer cells are HER2+) or chemotherapy [147,149,151].

HER2-positive (HER2+) breast cancers are characterized by positive expression of a molecule called HER2, which is classified as a protooncogene and encoded by a gene located at the long arm of human chromosome 17. It is noteworthy that increased expression of HER2 is found in many epithelial tumors. Due to the fact that HER2 belongs to the family of plasma membrane-bound receptor tyrosine kinases, this overexpression results in increased activity of this tyrosine kinase. It is active even in the absence of ligand, which is manifested, among others, by increased signaling promoting uncontrolled cell proliferation. HER2-positive cancers account for 15–20% of all breast cancers [152,153]. The presence of extra copies of the gene encoding HER2 often coexists with alterations in genes responsible for encoding proteins involved in proteolysis or angiogenesis. The repercussions of HER2 gene amplification include: higher risk of metastasis, worse clinical prognosis as well as shorter disease-free survival. Luminal HER2 and enriched HER2 were distinguished. The former (also called triple-positive) has expression of HER2, ER and PR, while its Ki-67 index has an intermediate value, which determines moderate proliferation. The latter has HER2 but no ER and PR, so no hormonal therapy is used. Despite the presence of the HER2 molecule, monoclonal antibody therapy is unsuccessful in almost 50% of patients, which, given the high value of the Ki-67 index noted in this cancer, and which indicates increased proliferation, implies a poor prognosis [154,155,156].

Triple-negative breast cancers, which account for about 15% of all breast cancers, owe their name to the fact that their cells lack ER, PR and HER2. They have a high Ki-67 index, which is evidence of increased proliferation. It has been proven that people with BRCA1 gene mutation have higher incidence of these cancers. Furthermore, they are found more often in women at a young age. This extremely aggressive (especially in African American women) and heterogeneous subtype of breast cancer is characterized by a very high risk of recurrence (local and systemic), which should be taken into consideration when introducing appropriate therapy. High incidence of early metastasis and recurrence determines a poorer prognosis [155,156,157].

### 3.2. Tumor Microenvironment

In recent years, our knowledge of the molecular basis of tumorigenesis in the breast has greatly expanded. It has been proven that cancer of this organ is accompanied by significant changes in the surrounding stroma. Many components of the so-called tumor microenvironment have been identified, as well as tumor-induced changes in the morphology and function of the ECM, immune cells, cytokines and growth factors and their receptors. Some of these changes are thought to facilitate tumor progression. However, alterations in the tumor microenvironment that inhibit tumor progression have also been identified. For example, the enrichment of cytotoxic T cells in the tumor microenvironment can be regarded as a tumor-induced modification while being anti-tumoral. Stromal cells in the breast cancer microenvironment are characterized by aberrant signaling pathways as well as molecular alterations that have prognostic significance for clinicians [159]. Breast cancer is now recognized as a highly heterogeneous (histologically and at the molecular level), genetically determined disease [148,151,152,154,155,156,157,160,161,162,163,164]. Both somatic and germline mutations are causative factors in tumorigenesis. It should be emphasized that certain mutations cause so-called hereditary tumor syndromes in patients, while repercussions of other mutations are certain morphological stages [160,161,162,163,164]. Links between genetic variation and pathological subtypes of breast cancer are the subject of research [164].

The breast cancer microenvironment can be considered at three main levels: local (intratumor), regional (in the breast) and distant (metastatic). Each of these levels contains: different cell types (leukocytes, fibroblasts, epithelial cells, adipocytes and myoepithelial cells), soluble factors (enzymes, growth factors, hormones and cytokines) as well as ECM with specific characteristics. Additionally, each has a different ion concentration (Ca^2+^ and H+) and oxygen content. The occurrence of interplay between components of the tumor microenvironment and breast cancer cells has been repeatedly studied and confirmed [159,165]. Some of the most important molecular players in the tumor microenvironment are T cells, which are the most abundant tumor infiltrating lymphocytes, as shown by experimental data. Recently, it has been demonstrated that T reg cells can promote breast cancer metastasis to bone by synthesizing and secreting receptor activator for nuclear factor kappa B ligand. Therefore, it has been suggested that the finding of multiple T reg cells in the tumor microenvironment worsens prognosis. The idea has been put forward that recruitment of these lymphocytes occurs as a result of prostaglandin E2 secretion by tumor cells. It has been reported that this effect is modulated by transforming growth factor β. The function of effector cells may be suppressed by the tumor via secretion of interleukin 10. The events described above contribute to the formation of the so-called immunosuppressive microenvironment, which is one of the key elements of the process referred to as immunoediting [159,166,167,168,169,170,171,172]. Tumor-associated macrophages (TAM), which originate in blood monocytes recruited at the tumor site via factors secreted by both neoplastic and stromal cells, also represent an important cell population in the breast cancer microenvironment. TAM exhibit a characteristic phenotype directed at promoting tumor growth, facilitating both angiogenesis (through producing VEGF) and tissue remodeling as well as suppressing adaptive immunity. Data collected so far indicate that high levels of TAM are associated with poor prognosis in breast cancer [159,173,174,175,176,177]. Tumor-associated stroma shows an abundance of immature dendritic cells (DC) with impaired capacity to stimulate antitumor immunity. These DC have the ability to promote tumor growth by enhancing endothelial cell migration and stimulating the production of proangiogenic factors. The cited DC activities disappear when these cells become mature. Moreover, infiltration of mature DC into primary tumor sites has been shown to reduce metastatic capacity, resulting in a better clinical outcome [159,178,179,180,181,182]. It is also worth highlighting that the role played by cancer-associated fibroblasts (CAF) in the breast cancer microenvironment, which are the source of many soluble factors (e.g., chemokines and growth factors), is not without significance. They are considered to be capable of enhancing tumor aggressiveness and facilitating metastasis. Compared to fibroblasts located in noncancerous tissues, CAF are characterized by significantly higher expression of genes related to morphogenesis and development. Furthermore, there are premises indicating that CAF might affect the transcriptional profile of breast cancer cells. These interactions may promote the formation and maintenance of a specific genetic–biochemical partnership to manage the microenvironment in such a way as to mutually facilitate access to nutrients. It is possible that the source of CAF is the bone marrow and their recruitment to the tumor microenvironment is accomplished by sending appropriate signals from tumor cells that are already present in this microenvironment, although it should be noted here that other concepts as to the provenance of CAF are also considered. While metalloproteinases produced by CAF appear to promote tumor invasion, other factors produced by these cells, such as caveolin-1 and podoplanin, which are associated with wound responses, have been linked with fewer nodal metastases [159,183,184,185,186,187,188,189,190]. It is worth mentioning that the heterogeneity of CAF has been recognized and the importance of four subsets (CAF-S1, CAF-S2, CAF-S3 and CAF-S4) of these cells, whose expression patterns in non-tumorigenic tissues and in breast cancer are different, has been described. CAF-S1 have been shown to be key immunosuppressive factors. They exhibit the ability to attract T lymphocytes and, moreover, to increase the survival of CD4+CD25+ T lymphocytes. Additionally, they facilitate the differentiation of these lymphocytes into CD25+FOXP3+ cells and stimulate T reg cells to block the proliferation of effector T cells [16,17]. The role of adipocytes in the tumor microenvironment is also important. In a healthy breast there are the following groups of adipocytes: adipose-derived stem cells, preadipocytes and mature adipocytes. Data collected so far indicate that in breast cancer tissue there are adipocytes with different characteristics (enhanced expression of adipokines and inflammatory factors, higher activity of matrix metalloproteinase, smaller size, increased expression of type VI collagen and decreased lipid content) from those found in the non-neoplastic tissue, therefore they are called cancer-associated adipocytes (CAA). CAA exhibit fibroblast-like phenotypes and possess senescent features (especially in obese people). They are located in the vicinity of tumor-transformed cells, with which, as it is presently assumed, they communicate chemically, inducing functional and phenotypic changes favoring tumor progression. Moreover, increased secretion by CAA of molecules, whose activity implies enhanced metastasis and tumor invasiveness, has been reported. The most important of these molecules are: interleukins (1β and 6), leptin, tumor necrosis factor α, parathyroid hormone-related protein, vascular endothelial growth factor and chemokine (C-C motif) ligands (2 and 5). The aforementioned communication at the CAA-tumor cell line also determines the metabolic reprogramming of CAA, which triggers their tumor-promoting potential. It has been discovered that exosomes can act as molecular linkers between CAA and breast cancer cells in enhancing tumorigenesis. Within the tumor microenvironment, exosomes carry onco-miRNA (miRNA-126, miRNA-144 and miRNA-155) from breast cells to adipocytes, leading to the conversion of the latter into CAA [18,19,20,21,22,23,24].

### 3.3. Essential Changes in Breast ECM during Carcinogenesis

These changes occur at every stage of carcinogenesis. In the non-tumorigenic breast, tissue microarchitecture is under precise multifactorial control [140,191].

Signaling for epithelial polarity is one key to the ECM role in tumor suppression. Underlying the disclosure of the tumor phenotype is the loss of this polarity, triggered by disruptions in cell–cell and cell–ECM interactions. If this polarity is restored, the process of carcinogenesis is inhibited [140,191,192,193,194]. The basement membrane can arrest nascent in situ carcinomas within its boundaries [140,195,196,197]. Its crossing by tumor-transformed cells is possible, among other reasons, because these cells are capable of disorganizing cell-to-cell and cell-to-ECM signaling pathways and can disrupt adhesion and migration. The role of cancer cell synthesis and secretion of enzymes that degrade ECM components is also emphasized [140,198,199,200,201,202,203]. Invasion of the basement membrane is usually temporally and spatially coordinated with increased protease synthesis, enhanced proteolysis, and abnormal turnover of matrix components via, among other things, endocytosis (e.g., laminin and its receptors) [140,204,205,206,207,208].

### 3.4. ECM Proteins in Breast Cancer

ECM proteins are mainly produced by myoepithelial cells. Therefore, changes in the synthesis of these proteins accompanying carcinogenesis are clearly visible in the mentioned cells. For instance, the loss of the ability of myoepithelial cells to synthesize laminin 111 and the inability of these cells to produce inhibitors of matrix-degrading proteases (such as maspin) have been observed. Conversion of carcinoma in situ to invasive breast cancer appears to be dependent on myoepithelial cell dysfunction [140,209,210,211,212]. Invasion and metastasis are preceded by an increase in collagen biosynthesis [140,213,214]. Upregulation of LOX enhances collagen cross-linking. The resulting stiffening of its structure is considered as one of the factors promoting metastasis [90,215,216,217,218]. Increased LOX activity has been shown to be induced by transforming growth factor β and hypoxia inducible factor [121,219,220]. Another enzyme involved in collagen metabolism (called prolyl hydroxylase) is also highly expressed in breast cancer tissues, which correlates with poor clinical outcomes. The subsequent reduction of collagen deposition due to silencing of the mentioned enzyme favors the reduction in metastasis (e.g., to lungs and lymph nodes), as well as decreasing the invasiveness of cancer cells [121,220,221,222]. There is speculation that increased breast density, which is generally associated with poor prognosis, is a consequence of increased collagen deposition and stiffness of the stromal matrix. This stiffness may further account for integrin clustering and increased activity of signaling pathways involving extracellular signal-regulated kinases (ERK). Due to degradation of the basement membrane, collagen type IV is decreased in breast cancer, while the number of fiber-forming collagens (types I, III, and V) is increased, which has been linked to a higher risk of invasion and malignancy. A collagen scaffold can be used by cancer cells during migration in order to facilitate this process [140,223,224].

It is now thought that many of the ECM proteins (e.g., periostin and tenascin C) are important components of the so-called pre-metastatic niche [135,215]. Periostin is produced by fibroblasts in tumor stroma [215,225,226]. It is important for normal skeletal and myocardial development and is also found in healthy tissues [135,227]. Its increased expression in tumors is usually associated with tissue stiffness-dependent facilitation of disease progression [135,217]. The mentioned increase in tissue stiffness is caused by an increase in LOX activity, which in turn results from an interaction between periostin and BMP-1. It is worth mentioning that LOX accumulates in the pre-metastatic niche and promotes recruitment of MMP-2 producing myeloid cells [135,228]. Moreover, periostin induces Wnt signaling by promoting recruitment of Wnt ligands, which also promotes metastasis formation [215,229]. Induction of periostin by transforming growth factor β3 facilitates breast cancer metastasis to the lung and survival of cancer cells in this organ, and increased plasma levels of this molecule have been linked to a higher risk of secondary breast cancer foci in the bone [135,229,230].

Tenascin C (TNC) assembles into a hexameric structure and is highly upregulated during tissue regeneration, because it participates in the formation and function of the provisional wound matrix [135,231]. TNC has been detected in both primary breast cancer and the invasive front of lung metastasis nodules. Both stromal and cancer cells express a significant amount of TNC. It is highly upregulated especially at invasive fronts [135,215,232]. TNC has the ability to modulate cancer stem cell signaling by enhancing expression of key regulators of the Wnt and Notch pathways, namely leucine-rich repeat-containing G protein-coupled receptor 5 (LGR5) and musashi homolog 1 (MSI1), respectively, which has been associated with an increased risk of recurrence (local and distant) [135,215,233]. TNC is one of six genes in a signature regulated by microRNA 335 in metastatic breast cancer [135,233]. At sites where tissues undergo remodeling, TNC typically coimmunoprecipitates with MMPs. Two of them (MMP-9 and MMP-13) are activated by TNC, which enhances breast cancer invasiveness. It has been reported that TNC expression limited only to the stroma is associated with better prognosis than its expression in both stromal and tumor cells. There is also evidence to suggest that TNC expression predicts poor 5-year survival in patients with breast cancer [135,234,235,236,237,238].

Osteonectin (ON) is a matricellular ECM protein that is nearly absent in normal mammary, however it is highly expressed in breast cancer [135,239]. This increased expression is mediated by β4 integrin, leading to increased invasiveness. ON is associated with basal, HER2+ and luminal B breast cancer subtypes while the luminal A subtype does not express this protein. ON regulates MMP-2 activity and facilitates metastasis to lung tissue [135,240,241,242]. There is some evidence suggesting an inverse correlation between ON and the estrogen receptor. The expression of ON in breast cancer is associated with poor metastasis-free survival as well as overall survival [135,243,244,245,246].

Thrombospondin 1 (THBS1) was originally detected in thrombocytes, but it also shows expression in osteoblasts, macrophages, fibroblasts, and tumor cells [135,247]. On the one hand, this molecule has been proven to inhibit the growth of primary tumors and block angiogenesis but, on the other hand, it has been noted to promote breast cancer metastasis to the lungs, which is most likely accomplished via activation of transforming growth factor β and stimulation of urokinase plasminogen activator. The expression of THBS1 in breast cancer associates with poor metastasis-free survival [135,248,249,250,251]. In the case of tumors that show neither estrogen receptor nor progesterone receptor expression, the increase in plasma THBS1 levels in diseased compared to healthy individuals may be of great clinical value owing to the fact that it can be considered as one of the markers of aggressiveness, since it has been noted that lymph node metastasis is much more frequent under the described conditions [135,252,253,254].

Osteopontin (OPN) is a phosphorylated glycoprotein that interacts with surface receptors including CD44 and several integrins, of which particular importance is commonly attributed to αvβ3 integrin on account of its participation in cell survival signaling. OPN occurs in bones and has a thrombin cleavage site. After cleavage, both fragments are recognized by integrin receptors. Thrombin cleavage of this molecule has been suggested to lead to an increase in OPN activity [135,255,256]. Overexpression of OPN results in increased tumor size, increased invasiveness, and promotes metastasis. Cancer cells with such an overexpression also have increased expression of urokinase plasminogen activator [135,257,258]. Most often overexpression of OPN occurs in stromal cells (lymphocytes and infiltrating macrophages), nevertheless this protein is also expressed by cancer cells directly and exists both as an immobilized part of the ECM and as a soluble factor circulating in the blood. OPN is expressed in node negative breast cancer. Its presence both in plasma and in tumor tissue may be a prognostic indicator of tumor aggressiveness. Indeed, low levels of this protein in blood plasma are associated with decreased metastatic spread and better overall survival [135,259,260,261,262]. Expression of OPN by orthotopically injected breast cancer cells is a necessary factor for the occurrence of bone marrow-derived stem cell mobilization, raising the possibility that this protein is not merely a passive biomarker [135,263].

Increased synthesis and enhanced deposition of fibronectin in tumor-affected tissues have also been found in human breast cancer [213,215,264]. Fibronectin has been detected in the stem cell niche. This molecule is considered as one of the indicators of EMT. For instance, it can promote EMT induced by transforming growth factor β. The effects of fibronectin on metastasis formation and EMT are mediated via the ERK/MAP kinase and Src kinase pathways [215,265,266]. It has been reported that ERK participates in one of the critical pathways in breast cancer progression. Studies to date suggest that binding of collagen type I to DDR stabilizes SNAIL1 (a transcription factor that promotes the repression of the adhesion molecule E-cadherin in order to regulate EMT) by stimulating ERK2 activity. Activated ERK2 can phosphorylate SNAIL. If this reaction occurs, SNAIL1 accumulates in the cell nucleus and subsequently promotes breast cancer invasion and enhances metastasis [215,267]. Abnormalities in the distribution of receptors for fibronectin on the surface of tumor cells were also highlighted. In general, fibronectin expression in breast cancer is associated with adverse clinical outcomes [268,269,270,271,272,273,274,275].

Associations of several laminin subtypes (111, 332, and 511) with tumorigenesis in the mammary gland have been established. Abnormal expression of laminin 111 or its loss, which are usually observed in the breast undergoing tumorigenesis, result in disturbed cell polarity. In view of the role of this laminin in the regulation of cell–cell adhesion, it is speculated that it has the ability to limit the spread of tumor cells [270,276,277,278,279]. Some studies have provided evidence that other laminins containing α4 subunits (such as laminin 332 and laminin 511) enhance cancer progression. Expression of laminin 332 accompanies aggressive breast cancer phenotype, whereas tumor-derived laminin 332 promotes anchorage-independent survival via interaction with integrin α6β4 receptors [270,280]. Interactions of laminin 332 with integrin α3 result in increased migration and invasion of tumor-transformed cells. Regarding laminin 511, it has been shown to have the ability to increase breast cancer invasiveness by promoting adhesion and migration of tumor cells. In a subpopulation of cells capable of self-renewal and tumor initiation, this laminin interacts with integrin α6β1 [270,281].

Regarding elastin, it is worth mentioning that elastosis, which results from an abnormal increase in expression of the components of elastin fibers and excessive degradation of normal elastic fibers, is a common feature in breast cancer. Elastosis increases with tumor progression. Ductal elastosis is particularly common in invasive cancer. Elastosis is recognized as a complex phenomenon resulting in both deposition of elastotic masses and local production of elastin fragments. These two manifestations must be distinguished within the matrix [270,282]. Elastin-derived peptides affect tumor cells and surrounding stroma. They promote invasion of this stroma and migration of cancer cells. These peptides also upregulate the expression of MMPs as well as facilitate chemotaxis, angiogenesis and elastase release. Moreover, they can prevent apoptosis [270,283,284]. The roles of the
selected ECM molecules in tumor microenvironment were shown in Figure 1.

**Figure 1 jcm-11-01250-f001:**
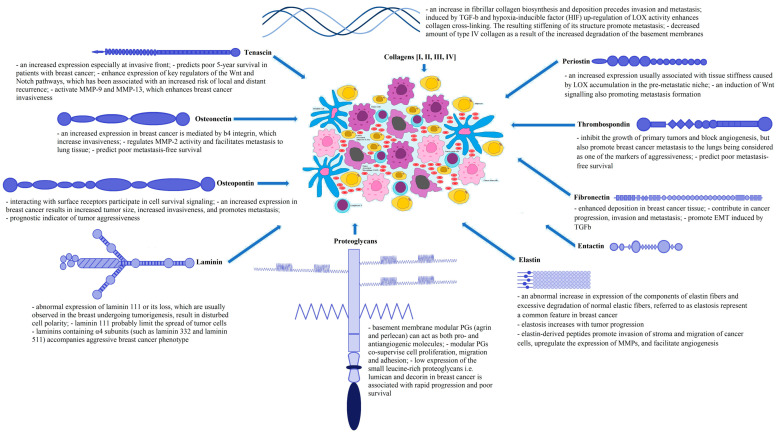
Chosen ECM molecules role in tumor microenvironment [10,12,15,65,66,90,121,135,140,213,214,215,216,217,218,219,220,225,226,228,229,232,234,235,236,237,238,240,241,242,243,244,245,246,248,249,250,251,255,256,264,265,266,267,270,276,277,278,279,281,282,283,284] modified.

### 3.5. Clinical Considerations

It seems obvious that clinical aspects of the discussed issues (e.g., the possibility of using ECM proteins as diagnostic markers in predicting the clinical course of the disease or the influence of specific anticancer therapies on the expression and function of the mentioned proteins in different subtypes of breast cancer) are of special interest for researchers dealing with breast cancer. In this chapter this topic will be discussed.

Data collected so far indicate that there is an association between the expression profiles of genes encoding specific ECM proteins and resistance of breast cancer cells (including metastatic ones) to chemotherapeutics. Increased expression of ON, POSTN, fibulin-1 and THBS2 has been shown to predispose stromal cells to show resistance to drugs such as cyclophosphamide (CPH), 5-fluorouracil (5-FU) and doxorubicin (DOXO) [285]. Moreover, it has been proven that THBS1 overexpression results in resistance to DOXO-mediated apoptosis of breast cancer cells (type I collagen seems to be involved in this effect), whereas OPN overexpression results in resistance to CPH-dependent apoptosis of these cells [286,287]. It has been suggested that, at least in some cases, the mere presence of a chemotherapeutic agent in the TME implies enhanced synthesis of specific ECM proteins. Interestingly, the present hypothesis has been confirmed in relation to DOXO, because in both in vitro and in vivo models it has been found that its presence in the breast cancer microenvironment induces an increase in the expression of laminin 111 and fibulin-1 [288,289]. Taking into account the above-described relationship between fibulin-1 and resistance of cancer cells to chemotherapeutic drugs, it can be speculated that this protein is a mediator of resistance induced by the presence of DOXO. It is worth noting that the effect of expression profiles of ECM proteins has been studied not only on the course of chemotherapy, but also on radiotherapy and hormone therapy. With regard to the last of the therapies mentioned in the previous sentence, it should be noted that the studies, which concerned the influence of ECM protein overexpression on tamoxifen therapy, provided the conclusions that increased levels of tenascin C, ON and fibronectin in TME predispose cancer cells to exhibit resistance to therapy with selective estrogen receptor modulators and, moreover, indicate poor prognosis [290]. With regard to laminins, one cannot help but mention the discovery that laminin 332 is responsible for the failure of anti-HER2 therapy in HER2-positive tumors. This effect is mediated by the following molecules: tetraspanin CD151 and two integrins (α6β4 and α3β1) [291]. It is also worth emphasizing that TNBC, which display mutant p53 and are characterized by enhanced angiogenesis and poor survival, lack laminins expression. ECM proteins are also important in the context of radiotherapy outcome as has been observed that fibronectin and laminins increase resistance to ionizing radiation in vitro [292]. The relationships described above encourage the search for possible therapeutic interventions, as described in Section 3.6. For some subtypes, correlations between the presence of mRNA of certain ECM proteins and clinical prognosis have been documented. It is important to note here that the presence of a particular protein in high concentration in a particular subtype does not imply at once that it is a diagnostic marker or prognostic indicator in that subtype. For instance, the concentration of mRNA for fibronectin in TNBC and HER2-positive, but it can be considered a prognostic marker only for those tumors that express ER and PR. In contrast, mRNA levels for ON are, admittedly, highest in the luminal A subtype. Nevertheless, it is not a prognostic indicator in luminal cancers, but in HER2-positive and basal subtypes. It follows from the above that caution should be exercised when interpreting the results of studies on concentrations of individual ECM proteins in a given subtype [293,294,295,296,297]. Moreover, the existence of different protein variants as a result of alternative splicing is not without significance. These variants usually undergo a process called isotype switching during tumorigenesis, which may affect the clinical effect of introduced therapies. As far as radiotherapy of breast cancer is concerned, it has been shown that those tumor cells which express higher level of splice form of fibronectin called ED-A, as well as its receptor—integrin α5β1, are more resistant to it. ED-A is a form that is particularly susceptible to polymerization and is associated with increased angiogenesis within the tumor. TNC also appears in the tumor-bearing breast in such isoforms, which are absent in the healthy body. It has been reported that the presence of these isoforms enhances invasiveness, in which matrix metalloproteinases are most likely involved [298,299,300,301,302,303,304,305,306,307]. Collagen type I, an essential determinant of stiffness in both healthy and cancerous breasts, has been proposed to link mammographically detected increased mammary gland density with increased breast cancer risk [308]. This role of collagen type I was confirmed in a mouse model, while in women a statistically significant correlation was observed between overexpression of genes encoding fibrillar collagens and increased degree of breast cancer invasiveness. The degree of collagen cross-linking was also found to influence invasiveness and prognosis. A loose structure of the network formed by this protein increases invasiveness, while a compact one reduces it. Sometimes, dense cross-linking of collagen fibers increases the local tissue density to such an extent that it can be detected by palpation of the breast [309,310,311]. It has also been observed that the density of chemotherapy-resistant tumors does not decrease after treatment (sometimes it even increases), whereas the density of chemotherapy-susceptible tumors decreases after treatment. It has been shown that collagen type III disorganizes the dense structure of collagen type I and furthermore impedes its formation, which is associated with a decrease in tumor aggressiveness. On the other hand, a decrease in collagen type III implies an increase in tumor invasiveness. A dense network of collagen fibers perpendicular to tumor border predicts invasiveness and poorer overall survival. Inhibition of lysine oxidase and blockade of transforming growth factor β result in a reduction in the stiffness and density of the collagen fiber network in the mammary gland, indicating this network as a potential therapeutic target for breast cancer treatment. In the context of the effectiveness of anticancer therapies, it is worth highlighting that a reduction (regardless of how this effect is achieved) in the density and stiffness of the collagen scaffold facilitates drug penetration. It has been observed that this is accompanied by a local reduction in fibrinogen accumulation and a decrease in the resistance of cancer cells to drug-induced apoptosis [312,313,314,315].

### 3.6. ECM Proteins as Targets for Anticancer Therapies

With regard to periostin (POSTN), it has been proven that neutralizing it with appropriate antibodies entailed a reduction in breast cancer metastasis to lung tissue. This is a promising result for future work in this area [135,316]. Furthermore, a POSTN-binding DNA aptamer has been shown to inhibit breast cancer growth and metastasis. It has been suggested that the use of such or similar aptamers may serve as a future therapeutic tool against those breast cancers that overexpress POSTN [317]. There are also high hopes for the effects of POSTN in the context of combating resistance to chemotherapeutics currently used to treat breast cancer. POSTN inhibition has been reported to overcome chemoresistance via reducing the expansion of mesenchymal tumor subpopulations in breast cancer. Knockdown of POSTN inhibited growth and invasion of mesenchymal tumor cells upon chemotherapy. Furthermore, chemotherapy upregulated cancer-specific variants of POSTN and application of a blocking antibody specifically targeting those variants overcame chemoresistance as well as halted disease progression in the absence of toxic effects [318].

Endostatin (an antiangiogenic factor that is a C-terminal fragment derived from collagen type XVIII) also appears to be a promising target for anticancer therapy. Endostatin has been shown to induce RAW264.7 phenotype polarization to M1 in vitro. There have been suggestions that it may inhibit breast cancer growth in mice in vivo via the regulation of polarization of TAM. Macrophage polarization is the process of differentiation of M0 macrophages into M1 or M2, in which these cells, due to the expression of various surface markers, show different functions in response to activating factors from the microenvironment. Macrophages with the M1 phenotype are pro-inflammatory cells with anti-tumor functions, and M2 macrophages have a tumor-promoting effect. It is suspected that this occurs by shifting the polarity of TAM from the M2-like to M1-like functional phenotype or by increasing the M1-like TAM via specific inhibition of M2 polarity. In addition, data collected so far indicate that the combination of chemotherapy with endostatin administration is characterized by higher efficacy than the implementation of chemotherapy alone. Based on these reports, it can be assumed that in the future the above-mentioned combination therapy may be a valuable option in the treatment of breast cancer. It seems, however, that further research is needed in this area owing to the fact that endostatin gene variation may be relevant in this regard [319,320,321,322,323,324,325,326,327,328,329].

Currently, in breast cancer research, three-dimensional (3D) in vitro models are used, in which it is possible to recreate the interactions between cancer cells and the extracellular matrix, as well as the relationship between cancer cells and stromal cells [330].

Interesting observations were provided by the studies conducted by Berger et al. [331], who analyzed the mechanism by which the stiffness of the substrate may influence the invasive behavior of breast cancer cells. Increasing stiffness from low to high (2 to 12 kPa) led to a switch from proteolytically independent invasion to a proteolytically dependent phenotype. The authors stated that cells in high stiffness had increased expression of Mena, an invadopodia protein associated with metastasis in breast cancer, as a result of EGFR and PLCγ1 activation. The results obtained provide important insight into the role of matrix stiffness, composition and organization in promoting cancer invasion [331]. The research conducted by Han et al. [332] showed an important role of spatiotemporal coordination of cellular physical properties in tissue organization and disease progression. According to the authors, using the multicellular model of the breast cancer organoid, we map the spatial and temporal evolution of the positions, movements and physical features of individual cells in three dimensions [332].

## 4. Conclusions

The relationships between ECM molecules and cancer development presented in this article show a significant relationship between the structure and function of the breast ECM and the interaction of many molecules both in physiological and pathological conditions. It is clear that any ECM reorganization in the breast must be under the strict and coordinated control of the organism. Disruption of cell–cell and cell–ECM interactions may lead to the development of a neoplastic process.

Moreover, the multitude and variety of interrelationships between the molecules that make up the tumor microenvironment makes it an important element, without understanding of which modern oncology will not be able to cope with many clinical challenges. Last but not least, it seems evident that as the understanding of the role of ECM proteins in breast cancer advances, there is a growing desire to put this knowledge into practice in the development and implementation of less toxic and more effective anti-cancer therapies.

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
