# Peer review of "The Role of Extracellular Matrix Proteins in Breast Cancer"

_jcm, 2022, doi:10.3390/jcm11051250_

Round 1

Reviewer 1 Report

In this review article, Lepucki et. al. has discussed the relevance of extracellular matrix proteins in breast cancer. This is a relevant and informative review article but can be significantly improved by the addition of a table and a figure. The manuscript should be rechecked for English structures and non-scientific vocabulary can be excluded. Additional comments are below:-

  1. Authors can add other components to the abstract. For example ECM and malignancy, ECM, and immune cells.
  2. In the abstract authors should mention other components of the extracellular matrix too, along with proteins.
  3. The authors can discuss the incidence and global burden (numbers) of breast cancer in the introduction.
  4. For the readers, authors can add a table, with 3 columns – biomolecule of ECM, brief function, relevant reference.
  5. For more emphasis, a figure should be added of ECM and its perturbation in the tumor microenvironment.
  6. In conclusion: ‘The cited and described in this manuscript relationships joining” should be re-phrased.
  7. “may confirm” should be replaced, the sentence can be restructured.
  8. “Once the body loses its ability to supervise it, tumorigenesis can begin.” The sentence can be restructured.
  9. “RAW264.7 phenotype polarization to M1 in vitro.” Authors can discuss macrophage polarization briefly.
  10. “Last but not least, it seems clear that as progress is made in understanding the role….” The sentence needs to be improved.
  11. “Many components of the so-called tumor mi..” Line 439 can be improved.
  12. Authors can add recent interesting papers :
  13. Berger AJ, Renner CM, Hale I, Yang X, Ponik SM, Weisman PS, Masters KS, Kreeger PK. Scaffold stiffness influences breast cancer cell invasion via EGFR-linked Mena upregulation and matrix remodeling. Matrix Biology. 2020 Jan 1;85:80-93.
  14. Authors can discuss the potential of 3D cultures in increasing our understanding of ECM interaction. For example: “Huerta‑Reyes M, Aguilar‑Rojas A. Three‑dimensional models to study breast cancer. International Journal of Oncology. 2021 Mar 1.
  15. “Han YL, Pegoraro AF, Li H, Li K, Yuan Y, Xu G, Gu Z, Sun J, Hao Y, Gupta SK, Li Y. Cell swelling, softening and invasion in a three-dimensional breast cancer model. Nature physics. 2020 Jan;16(1):101-8.”

Author Response

Answers to the comments of the Reviewer 1

We agree with the opinion of Reviewer 1. Detailed responses are presented below.

  1. “Authors can add other components to the abstract. For example ECM and malignancy, ECM, and immune cells”
  1. “In the abstract authors should mention other components of the extracellular matrix too, along with proteins”

According to the Reviewer’s advice the Abstract has been revised to add content on the link between ECM, immune cells and cancer. The ECM composition has been also described in more detail. Revised summary is presented below:

Abstract: The extracellular matrix is a structure composed of many molecules, including fibrillar (types I, II, III, V, XI, XXIV, XXVII) and non-fibrillar collagens (mainly basement membrane collagens: types IV, VIII, X), non-collagenous glycoproteins (elastin, laminin, fibronectin, thrombospondin, tenascin, osteopontin, osteonectin, entactin, periostin) embedded in a gel of negatively charged water-retaining glycosaminoglycans (GAGs) such as non-slufated hyaluronic acid (HA) and sulfated GAGs which are linked to a core protein to form proteoglycans (PGs). This highly dynamic molecular network provides critical biochemical and biomechanical cues that mediate the cell-cell and cell-matrix interactions, influence cell growth, migration and differentiation and serve as a reservoir of cytokines and growth factors’ action. The breakdown of normal ECM and its replacement with tumor ECM modulate the tumor microenvironment (TME) composition and is an essential part of tumourigenesis and metastasis, acting as key driver for malignant progression. Abnormal ECM also deregulate behavior of stromal cells as well as facilitate tumor-associated angiogenesis and inflammation. Thus, tumor matrix modulates each of the classically defined hallmarks of cancer promoting the growth, survival and invasion of the cancer.  Moreover, various ECM-derived components modulate the immune response affecting T cells, tumor associated macrophages (TAM), dendritic cells and cancer associated fibroblasts (CAF). This review article considers the role that extracellular matrix play in breast cancer. Determining the detailed connections between the ECM and cellular processes has helped to identify novel disease markers and therapeutic targets.

  1. “The authors can discuss the incidence and global burden (numbers) of breast cancer in the introduction”.

According to the Reviewer’s recommendation, more detailed information about the incidence and global burden of breast cancer has been added into the  Introduction section (p. Introduction – Anatomy of the breast, lines: 37-40]

 Female breast cancer is the leading cause of global cancer incidence, with an estimated 2.3 million women diagnosed with breast cancer and 685 000 deaths globally. At the end of 2020, there were 7.8 million women who had been diagnosed with breast cancer in the last 5 years, making it the most common cancer in the world.

  1. “For the readers, authors can add a table, with 3 columns – biomolecule of ECM, brief function, relevant reference”

As per the Reviewer’s advice, summarized information about the ECM biomolecules and their role in breast cancer has been added into the Table 1.

ECM biomolecule

Function in breast cancer

References

fiber-forming collagens (types I, II, III)

- an increase in fibrillar collagen biosynthesis and deposition precedes invasion and metastasis
- induced by TGF-b and hypoxia-inducible factor (HIF) up-regulation of LOX activity enhances collagen cross-linking. The resulting stiffening of its structure promote metastasis;

[140,213,214]

[90, 121,215–220]

network-forming collagens (type IV basement membrane collagen)

- decreased amount of type IV collagen as a result of the increased degradation of the basement membranes

[140]

periostin

- an increased expression usually associated with tissue stiffness caused by LOX accumulation in the pre-metastatic niche
- an induction of Wnt signalling also promoting metastasis formation

[215,225,226,228]

[215,229]

tenascin C (TNC)

- an increased expression especially at invasive front
- predicts poor 5-year survival in patients with breast cancer
- enhance expression of key regulators of the Wnt and Notch pathways, which has been associated with an increased risk of local and distant recurrence
- activate MMP-9 and MMP-13, which enhances breast cancer invasiveness

[135,215,232]

[135,234–238]

[135,215,233]

entactin

- binds to laminin, collagen IV, fibrinogen, and fibronectin playing an important role in the assembly and properties of extracellular matrix
- a component of basement membranes, the loss of which is a key stage in malignant progression
- take part in controlling cell adhesion, which may be important for its role in tumorigenesis

[133,134]

osteonectin (ON)

- an increased expression in breast cancer is mediated by b4 integrin, which increase invasiveness
- regulates MMP-2 activity and facilitates metastasis to lung tissue
- predict poor metastasis-free survival

[135,239]

[135,240–242]

[243-246]

trombospondin 1 (THBS1)

- inhibit the growth of primary tumors and block angiogenesis, but also promote breast cancer metastasis to the lungs being considered as one of the markers of aggressiveness

- predict poor metastasis-free survival

[135,248–251]

osteopontin (OPN)

- interacting with surface receptors participate in cell survival signaling
- an increased expression in breast cancer results in increased tumor size, increased invasiveness, and promotes metastasis
- prognostic indicator of tumor aggressiveness

[135,255,256]

fibronectin (FN)

- enhanced deposition in breast cancer tissue

- contribute in cancer progression, invasion and metastasis

- promote EMT induced by TGFb

[213,215,264]

[215,267]

[215,265,266]

proteoglycans (PGs)

- basement membrane modular PGs (agrin and perlecan) can act as both pro- and antiangiogenic molecules
- modular PGs co-supervise cell proliferation, migration and adhesion
- low expression of the small leucine-rich proteoglycans i.e. lumican and decorin in breast cancer is associated with rapid progression and poor survival

[10,12,15]

[65,66]

[235]

laminin (LN)

- abnormal expression of laminin 111 or its loss, which are usually observed in the breast undergoing tumorigenesis, result in disturbed cell polarity
- laminin 111 probably limit the spread of tumor cells
- laminins containing α4 subunits (such as laminin 332 and laminin 511) accompanies aggressive breast cancer phenotype

[270,276-279, 281]

elastin

- an abnormal increase in expression of the components of elastin fibers and excessive degradation of normal elastic fibers, referred to as elastosis represent a common feature in breast cancer
- elastosis increases with tumor progression
- elastin-derived peptides promote invasion of stroma and migration of cancer cells, upregulate the expression of MMPs, and facilitate angiogenesis

[270,282-284]

  1. “For more emphasis, a figure should be added of ECM and its perturbation in the tumor microenvironment”.

According to the Reviewer’s recommendation, a figure on ECM and its perturbation in the tumor microenvironment has been added into the 3.4. ECM proteins in breast cancer section (line: 743]

  1. In conclusion: ‘The cited and described in this manuscript relationships joining” should be re-phrased.

The conclusion was corrected as suggested by the Reviewer (lines 794-809).

  1. “may confirm” should be replaced, the sentence can be restructured.

Corrected according to the Reviewer’s suggestion.

  1. “Once the body loses its ability to supervise it, tumorigenesis can begin.” The sentence can be restructured.

Corrected according to the Reviewer’s suggestion.

  1. “RAW264.7 phenotype polarization to M1 in vitro.” Authors can discuss macrophage polarization briefly.

As suggested by the Reviewer, we have included the definition of macrophage polarization in the text.

  1. “Last but not least, it seems clear that as progress is made in understanding the role….” The sentence needs to be improved.

Corrected according to the Reviewer’s suggestion.

  1. “Many components of the so-called tumor mi..” Line 439 can be improved.

Corrected according to the Reviewer’s suggestion.

  1. Authors can add recent interesting papers :

Corrected according to the Reviewer’s suggestion, taking into account the indicated publications.

  1. Berger AJ, Renner CM, Hale I, Yang X, Ponik SM, Weisman PS, Masters KS, Kreeger PK. Scaffold stiffness influences breast cancer cell invasion via EGFR-linked Mena upregulation and matrix remodeling. Matrix Biology. 2020 Jan 1;85:80-93.
  2. Authors can discuss the potential of 3D cultures in increasing our understanding of ECM interaction. For example: “Huerta Reyes M, Aguilar Rojas A. Three dimensional models to study breast cancer. International Journal of Oncology. 2021 Mar 1.
  3. “Han YL, Pegoraro AF, Li H, Li K, Yuan Y, Xu G, Gu Z, Sun J, Hao Y, Gupta SK, Li Y. Cell swelling, softening and invasion in a three-dimensional breast cancer model. Nature physics. 2020 Jan;16(1):101-8.”

Reviewer 2 Report

The authors describe the different proteins present in the extracellular matrix and their importance in the context of breast cancer. The field is raising awareness in the scientific community. The manuscript is well written, some modifications could be added to increase its comprehensiveness and readability.

  1. General remarks
    • An illustration would be appreciated to summarize the different aspects cover by the review. It would be good to at least illustrate the spatial distribution of the different mentioned proteins in the ECM.
    • The impact of breast cancer treatments on the proteins from the ECM or the TME in general is not clearly mentioned, maybe this could be described.
    • Adipocytes are mentioned in the introduction of section 3.1, but no more described after. It would be good to have a word for them in the TME section for instance.
    • The plan could be adapted as the label “an introduction” in section 3 is followed by conclusion section 4, given the feeling that the manuscript does not have more than an introduction
    • As mentioned in the abstract the ECM is made of many molecules, the review could acknowledge its limitations by opening to other kinds of molecules such as the lipids that are nor covered here but remain important in the context of breast cancer.
    • Breast cancer histology is never clearly discussed, while lobular breast cancer is defined by strongly discohesive tumor cells with a greater ECM remodeling as compared to breast cancer of non-special types.
  2. Section 2:
    • The link with cancer is not always clear, systematically adding some examples that illustrate the relevance in cancer would help the reader like it has been done for some subsections 2.5/2.6/2.7/2.9. When not possible, a reference to section 3.3 could be made to mention the reader that the role in cancer will be described later in the manuscript.
  3. Section 3:
    • In introduction: the molecular subtypes should be introduced here (ER/PR/HER2) given their importance and the fact you are using them in 3.3
    • Overall, in the section: maybe precise in which subtypes the findings have been made when possible.
    • Line 427. The sentence is misleading, in the metastatic setting, the cure is not achievable for now. The 20% between brackets refers to (according to the reference) the 5-year survival rates.
    • Line 442: As a comment: Is the anti-tumor induced modification always pro-tumoral? For instance, cytotoxic T cells enrichment in the TME could be seen as a tumor induced modification while being anti-tumoral.
    • Line 446. I would rather refer to histology and molecular levels of heterogeneity, pathology and imaging being a means to assess these layers.
    • Line 483: About the CAF, their heterogeneity should be mentioned (i.e. with Costa A, Kieffer Y, Scholer-Dahirel A, Pelon F, Bourachot B, Cardon M, et al. Fibroblast Heterogeneity and Immunosuppressive Environment in Human Breast Cancer. Cancer Cell. 2018;33:463-479.e10.)

Author Response

Answers to the comments of the Reviewer 2

We agree with the opinion of Reviewer 2. Detailed responses are presented below.

  1. Adipocytes are mentioned in the introduction of section 3.1, but no more described after. It would be good to have a word for them in the TME section for instance.

Ad. 1)

The role of adipocytes in the tumor microenvironment is also important. In a healthy breast there are the following groups of adipocytes: adipose-derived stem cells, preadipocytes and mature adipocytes. Data collected so far indicate that in breast cancer tissue there are adipocytes with different characteristics (enhanced expression of adipokines and inflammatory factors, higher activity of matrix metalloproteinase, smaller size, increased expression of type VI collagen and decreased lipid content) from those found in the non-neoplastic tissue, therefore they are called cancer-associated adipocytes (CAA). CAA exhibit fibroblast-like phenotypes and possess senescent features (especially in obese people). They are located in the vicinity of tumor-transformed cells, with which, as it is presently assumed, they communicate chemically, inducing functional and phenotypic changes favoring tumor progression. Moreover, increased secretion by CAA of molecules, whose activity implies enhanced metastasis and tumor invasiveness, has been reported. The most important of these molecules are: interleukins (1β and 6), leptin, tumor necrosis factor α, parathyroid hormone-related protein, vascular endothelial growth factor and chemokine (C-C motif) ligands (2 and 5). The aforementioned communication at the CAA-tumor cell line also determines the metabolic reprogramming of CAA, which triggers their tumor-promoting potential. It has been discovered that exosomes can act as molecular linkers between CAA and breast cancer cells in enhancing tumorigenesis. Within the tumor microenvironment, exosomes carry onco-miRNA (miRNA-126, miRNA-144 and miRNA-155) from breast cells to adipocytes, leading to the conversion of the latter into CAA [18-24].

  1. In introduction: the molecular subtypes should be introduced here (ER/PR/HER2) given their importance and the fact you are using them in 3.3

Ad. 2)

3.1. Molecular subtypes of breast cancer

Based on the expression profile of specific genes, supported by immunohistochemical assays, the following molecular subtypes of breast cancer were identified: luminal A, luminal B, luminal HER2, enriched HER2 and triple-negative.

Luminal A subtype is the most common (accounting for nearly 50% of all newly diagnosed breast cancer cases) subtype and also the least aggressive [2]. It shares some features with luminal breast epithelial cells, namely, it manifests high expression of cytokeratins (7, 8, 18 and 19). Besides, the expression of proliferation-stimulating genes is low in this subtype (low Ki-67 index), while the prognosis is very good. Lymph node involvement is rare, and the clinical course is relatively benign. It expresses estrogen receptor (ER) and progesterone receptor (PR), however the expression of human epidermal growth factor receptor 2 (HER2) is very low in this subtype. This constellation of properties makes luminal A subtype susceptible to hormonal therapy (using aromatase inhibitors or selective estrogen receptor modulators) [2-5].

Luminal subtype B accounts for 20-30% of invasive breast cancer cases and it is worth noting that most breast cancers genetically determined by BRCA2 gene mutation belong to this subtype [6]. This subtype is characterized by: high expression of cytokeratins, intermediate prognosis, higher risk of local recurrence after treatment (than in the case of subtype A). It is assumed that higher (than in subtype A) expression of Ki67, cyclin E1 and nuclease sensitive element binding protein 1, indicating increased proliferation, implies worse prognosis than in subtype A [4,6,7]. It seems that increased signaling via pathways involving Src and PI3K kinases is also significant in this regard. This subtype is usually treated as the most aggressive form of hormone-dependent breast cancer. In addition to hormone therapy, this subtype usually requires additional treatment options: targeted therapy (if the cancer cells are HER2+) or chemotherapy [8-10].

HER2-positive (HER2+) breast cancers are characterized by positive expression of a molecule called HER2, which is classified as a protooncogene and encoded by a gene located at the long arm of human chromosome 17. It is noteworthy that increased expression of HER2 is found in many epithelial tumors. Due to the fact that HER2 belongs to the family of plasma membrane-bound receptor tyrosine kinases, this overexpression results in increased activity of this tyrosine kinase. It is active even in the absence of ligand, which is manifested, among others, by increased signaling promoting uncontrolled cell proliferation. HER2-positive cancers account for 15-20% of all breast cancers [4,11]. The presence of extra copies of the gene encoding HER2 often coexists with alterations in genes responsible for encoding proteins involved in proteolysis or angiogenesis. The repercussions of HER2 gene amplification include: higher risk of metastasis, worse clinical prognosis as well as shorter disease-free survival. Luminal HER2 and enriched HER2 were distinguished. The former (also called triple-positive) has expression of HER2, ER and PR, while its Ki-67 index has an intermediate value, which determines moderate proliferation. The latter has HER2 but no ER and PR, so no hormonal therapy is used. Despite the presence of the HER2 molecule, monoclonal antibody therapy is unsuccessful in almost 50% of patients, which, given the high value of the Ki-67 index noted in this cancer, and which indicates increased proliferation, implies a poor prognosis [4,6,12-15].

Triple-negative breast cancers, which account for about 15% of all breast cancers, owe their name to the fact that their cells lack ER, PR and HER2. They have a high Ki-67 index, which is evidence of increased proliferation. It has been proven that people with BRCA1 gene mutation have higher incidence of these cancers. Furthermore, they are found more often in women at a young age. This extremely aggressive (especially in African American women) and heterogeneous subtype of breast cancer is characterized by a very high risk of recurrence (local and systemic), which should be taken into consideration when introducing appropriate therapy. High incidence of early metastasis and recurrence determines a poorer prognosis [4,13,15].

  1. Line 427. The sentence is misleading, in the metastatic setting, the cure is not achievable for now. The 20% between brackets refers to (according to the reference) the 5-year survival rates.

Ad.3)

In the absence of any metastases, the cure rate of patients sometimes reaches 90%, whereas in the metastatic setting, the cure is not achievable for now. It is believed that in such cases the long-term survival depends mainly on the organs to which the metastases occur, as well as the extent and speed with which it occurs [157]. Early diagnosis facilitates treatment and is associated with a better prognosis [148].

  1. Line 442: As a comment: Is the anti-tumor induced modification always pro-tumoral? For instance, cytotoxic T cells enrichment in the TME could be seen as a tumor induced modification while being anti-tumoral.

Ad. 4)

Many components of the so-called tumor microenvironment have been identified, as well as tumor-induced changes in the morphology and function of the ECM, immune cells, cytokines and growth factors and their receptors. Some of these changes are thought to facilitate tumor progression. However, alterations in the tumor microenvironment that inhibit tumor progression have also been identified. For example, the enrichment of cytotoxic T cells in the tumor microenvironment can be regarded as a tumor-induced modification while being anti-tumoral.

  1. Line 446. I would rather refer to histology and molecular levels of heterogeneity, pathology and imaging being a means to assess these layers.

Ad. 5)

Breast cancer is now recognized as a highly heterogeneous (histologically and at the molecular level), genetically determined disease [148,151,152,154-157,160–164].

  1. Line 483: About the CAF, their heterogeneity should be mentioned (i.e. with Costa A, Kieffer Y, Scholer-Dahirel A, Pelon F, Bourachot B, Cardon M, et al. Fibroblast Heterogeneity and Immunosuppressive Environment in Human Breast Cancer. Cancer Cell. 2018;33:463-479.e10.).

Ad. 6)

It is worth mentioning that the heterogeneity of CAF has been recognized and the importance of four subsets (CAF-S1, CAF-S2, CAF-S3 and CAF-S4) of these cells, whose expression patterns in non-tumorigenic tissues and in breast cancer are different, has been described. CAF-S1 have been shown to be key immunosuppressive factors. They exhibit the ability to attract T lymphocytes and, moreover, to increase the survival of CD4+CD25+ T lymphocytes. Additionally, they facilitate the differentiation of these lymphocytes into CD25+FOXP3+ cells and stimulate T reg cells to block the proliferation of effector T cells [16,17].

  1. The plan could be adapted as the label “an introduction” in section 3 is followed by conclusion section 4, given the feeling that the manuscript does not have more than an introduction.

Ad. 7)

The plan has been revised as recommended by the reviewer.

  1. The link with cancer is not always clear, systematically adding some examples that illustrate the relevance in cancer would help the reader like it has been done for some subsections 2.5/2.6/2.7/2.9. When not possible, a reference to section 3.3 could be made to mention the reader that the role in cancer will be described later in the manuscript.

Ad. 8)

This section focuses on describing the general properties of ECM proteins, while their role in breast cancer will be discussed in detail in Section 3.4.

Reviewer 3 Report

In the manuscript entitled “The role of extracellular matrix proteins in breast cancer”, Lepucki and the colleagues attempted to review a detailed biochemical approach to ECM proteins in breast cancer. I think this manuscript is a good, in-depth review of the vast ECM protein and TME. However, in this manuscript, the description of the chemical approach is major. It appears that this manuscript has a different focus from the aim pursued in the Journal of Clinical Medicines. 
Journal of Clinical Medicines is an international, peer-reviewed, open access journal on clinical and pre-clinical research, as stated on the journal homepage. For this manuscript to be of greater value than this journal, a review of clinical approaches to breast cancer, TME, and the role of ECM is needed. Despite the limitations that ECM-targeted therapeutics have not yet been developed, this manuscript lacks a description of the treatment or prognosis. Especially from a therapeutic point of view, this manuscript contains only one chapter "3.4. ECM proteins as targets for anticancer therapies". For submission to this journal, it is recommended that this manuscript include a description of the preclinical or clinical trials with sufficient consideration. 
As described in this manuscript, breast cancer is considered a disease composed of various subtypes due to genetic mutations and the like. In particular, different approaches are needed for each subtype of estrogen, progesterone, and HER2. In this manuscript, a description of the role of ECM in different subtypes of breast cancer is lacking. Therefore, overall, this manuscript needs to add sufficient consideration to the clinical point of view of breast cancer.  

Author Response

Answers to the comments of the Reviewer 3

We agree with the opinion of Reviewer 3. Detailed responses are presented below.

“In the manuscript entitled “The role of extracellular matrix proteins in breast cancer”, Lepucki and the colleagues attempted to review a detailed biochemical approach to ECM proteins in breast cancer. I think this manuscript is a good, in-depth review of the vast ECM protein and TME. However, in this manuscript, the description of the chemical approach is major. It appears that this manuscript has a different focus from the aim pursued in the Journal of Clinical Medicines.

Journal of Clinical Medicines is an international, peer-reviewed, open access journal on clinical and pre-clinical research, as stated on the journal homepage. For this manuscript to be of greater value than this journal, a review of clinical approaches to breast cancer, TME, and the role of ECM is needed. Despite the limitations that ECM-targeted therapeutics have not yet been developed, this manuscript lacks a description of the treatment or prognosis. Especially from a therapeutic point of view, this manuscript contains only one chapter "3.4. ECM proteins as targets for anticancer therapies". For submission to this journal, it is recommended that this manuscript include a description of the preclinical or clinical trials with sufficient consideration.

As described in this manuscript, breast cancer is considered a disease composed of various subtypes due to genetic mutations and the like. In particular, different approaches are needed for each subtype of estrogen, progesterone, and HER2. In this manuscript, a description of the role of ECM in different subtypes of breast cancer is lacking. Therefore, overall, this manuscript needs to add sufficient consideration to the clinical point of view of breast cancer.”

According to current state of knowledge – Breast cancer is characterized with enormous heterogeneity. It can be classified into four subtypes: Luminal A, luminal B, human epidermal growth factor receptor 2 (HER2) -positive and basal-like, according to clinically and biologically relevant molecular features. Molecular classification of breast cancers providing sufficient prognostic and predictive power and thus biomarkers  have been actively pursued in biomedical researches. Basal-like breast cancer is predominantly triple-negative (i.e. no expression of estrogen receptor (ER), progesterone receptor (PR) or HER2). In HER2-positive breast cancer, the HER2 gene is overexpressed. The basal-like and HER2-positive subtypes are associated with a poor prognosis, and the luminal A subtype was identified as having the more favorable clinical outcome [Wang J, Du Q, Li C. Bioinformatics analysis of gene expression profiles to identify causal genes in luminal B2 breast cancer. Oncol Lett. 2017 Dec;14(6):7880-7888. doi: 10.3892/ol.2017.7256. Epub 2017 Oct 230] – during our next step of experimental procedures we (our scientific team) not only want to examine the influence of different treatment methods on connective tissue but also we would like to try to implement innovative/alternative diagnostic algorithms.

Futhermore taking into acccount current state of knowledge – Scientific research confirms that the stiffness of the extracellular matrix correlates with the aggressiveness of the tumor, which is manifested by a greater tendency to local invasion and metastasis. This is associated with a worse prognosis of the patient. Experimental studies conducted on mice also confirmed that the mechanical properties of neoplastic tissue may also affect tumor response to systemic treatment (Takai, P. Lu, K.; Weaver, VM.; Werb, Z. Extracellular matrix degradation and remodeling in development and disease. Cold Spring Harb Perspect Biol. 2011; 3(12). The mechanical properties of the extracellular matrix may differ from one tumor type to another. In the most aggressive breast cancers, in the basal and non-luminal types, the extracellular matrix is characterized by the greatest stiffness and heterogeneity compared to less aggressive cancers (Luminal A and B types) (Acerbi). Moreover, in luminal breast cancers, apart from the lower stiffness of the extracellular matrix, there is also less stimulation of the immune system confirmed by smaller infiltration of immune cells (Acerbi I, Cassereau L, Dean I, Shi Q, Au A, Park C, Chen YY, Liphardt J, Hwang ES, Weaver VM. Human breast cancer invasion and aggression correlates with ECM stiffening and immune cell infiltration. Integr Biol (Camb). 2015 Oct;7(10):1120-34. doi: 10.1039/c5ib00040h) – in order to implement innovative/alternative diagnostic algorithms we would also like to continue our cooperation with Silesian University Specialsts – professor Koprowski. During our previous cooperstion with professor Koprowski we join our diagnostic, biochemic, pharmaceutic and bioinformatic knowledge in the field of Diabetes Mellitus experiments.

Clinical Trial Molecules. 2017 Jul 30;22(8):1274. doi: 10.3390/molecules22081274.

Adiponectin, Leptin, and Leptin Receptor in Obese Patients with Type 2 Diabetes Treated with Insulin Detemir

Paweł Olczyk 1, Robert Koprowski 2, Katarzyna Komosinska-Vassev 3, Agnieszka Jura-Półtorak 4, Katarzyna Winsz-Szczotka 5, Kornelia Kuźnik-Trocha 6, Łukasz Mencner 7, Alicja Telega 8, Diana Ivanova 9, Krystyna Olczyk 10

Affiliations expand

PMID: 28758947 PMCID: PMC6152287 DOI: 10.3390/molecules22081274

Free PMC article

https://pubmed.ncbi.nlm.nih.gov/28758947/

Round 2

Reviewer 1 Report

The authors have significantly improved the manuscript.

Author Response

Answers to the comments of the Reviewer 1

We would like to thank the Honorable Reviewer 1 for a detailed and extremely helpful review, for valuable comments, which we then introduced to the texts of the publication.

Last but not least, we would also like to thank the Reviewer 1 for accepting the publication submitted after the corrections.

Reviewer 3 Report

Lepucki et al. did a very good review of the role of ECM proteins in breast cancer. This manuscript will provide readers with a broad understanding of the role of the ECM in breast cancer. In particular, the summary figures are very intuitive and provide a good overview of the role of ECM proteins in breast cancer. However, as with the last review, my concern is that this manuscript does not contain any clinical focus or clinical considerations for breast cancer. In this revision manuscript, a review of subtypes of breast cancer is added to the previous manuscript. In my last review, I suggested that a review of the role of ECM proteins in each subtype of breast cancer is needed. There is absolutely no need for a review per se, just the subtypes of breast cancer described in the revision manuscript. In this manuscript, clinically important parts (for example, studies as a diagnostic value or biomarker in the role of ECM protein, or research on target therapeutics and clinical trials, or prognostic studies through ECM protein research, etc.) should be reviewed. This revision manuscript does not add any of these clinical aspects, and lacks a response to the last review. It seems that the authors lack an understanding of the clinical part of breast cancer. This manuscript will be more appreciated in the field of biochemistry.

Author Response

Answers to the comments of the Reviewer 3

We agree with the opinion of Reviewer 3. Detailed responses are presented below.

“In the manuscript entitled “The role of extracellular matrix proteins in breast cancer”, Lepucki and the colleagues attempted to review a detailed biochemical approach to ECM proteins in breast cancer. I think this manuscript is a good, in-depth review of the vast ECM protein and TME. However, in this manuscript, the description of the chemical approach is major. It appears that this manuscript has a different focus from the aim pursued in the Journal of Clinical Medicines.

Journal of Clinical Medicines is an international, peer-reviewed, open access journal on clinical and pre-clinical research, as stated on the journal homepage. For this manuscript to be of greater value than this journal, a review of clinical approaches to breast cancer, TME, and the role of ECM is needed. Despite the limitations that ECM-targeted therapeutics have not yet been developed, this manuscript lacks a description of the treatment or prognosis. Especially from a therapeutic point of view, this manuscript contains only one chapter "3.4. ECM proteins as targets for anticancer therapies". For submission to this journal, it is recommended that this manuscript include a description of the preclinical or clinical trials with sufficient consideration.

As described in this manuscript, breast cancer is considered a disease composed of various subtypes due to genetic mutations and the like. In particular, different approaches are needed for each subtype of estrogen, progesterone, and HER2. In this manuscript, a description of the role of ECM in different subtypes of breast cancer is lacking. Therefore, overall, this manuscript needs to add sufficient consideration to the clinical point of view of breast cancer.”

According to the Reviewer’s advice “the consideration to the clinical point of view of breast cancer” has been have been discussed in the manuscript, in “subsection 3.5. Clinical consideration (lines: 754 - 832)”:

3.5. Clinical considerations

It seems obvious that clinical aspects of the discussed issues (e.g. the possibility of using ECM proteins as diagnostic markers in predicting the clinical course of the disease or the influence of specific anticancer therapies on the expression and function of the mentioned proteins in different subtypes of breast cancer) are of special interest for researchers dealing with breast cancer. In this chapter this topic will be discussed.

Data collected so far indicate that there is an association between the expression profiles of genes encoding specific ECM proteins and resistance of breast cancer cells (including metastatic ones) to chemotherapeutics. Increased expression of ON, POSTN, fibulin-1 and THBS2 has been shown to predispose stromal cells to show resistance to drugs such as cyclophosphamide (CPH), 5-fluorouracil (5-FU) and doxorubicin (DOXO) [285]. Moreover, it has been proven that THBS1 overexpression results in resistance to DOXO-mediated apoptosis of breast cancer cells (type I collagen seems to be involved in this effect), whereas OPN overexpression results in resistance to CPH-dependent apoptosis of these cells [286,287]. It has been suggested that, at least in some cases, the mere presence of a chemotherapeutic agent in the TME implies enhanced synthesis of specific ECM proteins. Interestingly, the present hypothesis has been confirmed in relation to DOXO, because in both in vitro and in vivo models it has been found that its presence in the breast cancer microenvironment induces an increase in the expression of laminin 111 and fibulin-1 [288,289]. Taking into account the above-described relationship between fibulin-1 and resistance of cancer cells to chemotherapeutic drugs, it can be speculated that this protein is a mediator of resistance induced by the presence of DOXO. It is worth noting that the effect of expression profiles of ECM proteins has been studied not only on the course of chemotherapy, but also on radiotherapy and hormone therapy. With regard to the last of the therapies mentioned in the previous sentence, it should be noted that the studies, which concerned the influence of ECM protein overexpression on tamoxifen therapy, provided the conclusions that increased levels of tenascin C, ON and fibronectin in TME predispose cancer cells to exhibit resistance to therapy with selective estrogen receptor modulators and, moreover, indicate poor prognosis [290]. With regard to laminins, one cannot help but mention the discovery that laminin 332 is responsible for the failure of anti-HER2 therapy in HER2-positive tumors. This effect is mediated by the following molecules: tetraspanin CD151 and two integrins (α6β4 and α3β1) [291]. It is also worth emphasizing that TNBC, which display mutant p53 and are characterized by enhanced angiogenesis and poor survival, lack laminins expression. ECM proteins are also important in the context of radiotherapy outcome as it has been observed that fibronectin and laminins increase resistance to ionizing radiation in vitro [292]. The relationships described above encourage the search for possible therapeutic interventions, as described in section 3.6. For some subtypes, correlations between the presence of mRNA of certain ECM proteins and clinical prognosis have been documented. It is important to note here that the presence of a particular protein in high concentration in a particular subtype does not imply at once that it is a diagnostic marker or prognostic indicator in that subtype. For instance, the concentration of mRNA for fibronectin in TNBC and HER2-positive, but it can be considered a prognostic marker only for those tumors that express ER and PR. In contrast, mRNA levels for ON are, admittedly, highest in the luminal A subtype. Nevertheless, it is not a prognostic indicator in luminal cancers, but in HER2-positive and basal subtypes. It follows from the above that caution should be exercised when interpreting the results of studies on concentrations of individual ECM proteins in a given subtype [293-297]. Moreover, the existence of different protein variants as a result of alternative splicing is not without significance. These variants usually undergo a process called isotype switching during tumorigenesis, which may affect the clinical effect of introduced therapies. As far as radiotherapy of breast cancer is concerned, it has been shown that those tumor cells which express higher level of splice form of fibronectin called ED-A, as well as its receptor – integrin α5β1, are more resistant to it. ED-A is a form that is particularly susceptible to polymerization and is associated with increased angiogenesis within the tumor. TNC also appears in the tumor-bearing breast in such isoforms, which are absent in the healthy body. It has been reported that the presence of these isoforms enhances invasiveness, in which matrix metalloproteinases are most likely involved [298-307]. Collagen type I, an essential determinant of stiffness in both healthy and cancerous breasts, has been proposed to link mammographically detected increased mammary gland density with increased breast cancer risk [308]. This role of collagen type I was confirmed in a mouse model, while in women a statistically significant correlation was observed between overexpression of genes encoding fibrillar collagens and increased degree of breast cancer invasiveness. The degree of collagen cross-linking was also found to influence invasiveness and prognosis. A loose structure of the network formed by this protein increases invasiveness, while a compact one reduces it. Sometimes, dense cross-linking of collagen fibers increases the local tissue density to such an extent that it can be detected by palpation of the breast [309-311]. It has also been observed that the density of chemotherapy-resistant tumors does not decrease after treatment (sometimes it even increases), whereas the density of chemotherapy-susceptible tumors decreases after treatment. It has been shown that collagen type III disorganizes the dense structure of collagen type I and furthermore, impedes its formation, which is associated with a decrease in tumor aggressiveness. On the other hand, a decrease in collagen type III implies an increase in tumor invasiveness. A dense network of collagen fibers perpendicular to tumor border predicts invasiveness and poorer overall survival. Inhibition of lysine oxidase and blockade of transforming growth factor β result in a reduction in the stiffness and density of the collagen fiber network in the mammary gland, indicating this network as a potential therapeutic target for breast cancer treatment. In the context of the effectiveness of anticancer therapies, it is worth highlighting that a reduction (regardless of how this effect is achieved) in the density and stiffness of the collagen scaffold facilitates drug penetration. It has been observed that this is accompanied by a local reduction in fibrinogen accumulation and a decrease in the resistance of cancer cells to drug-induced apoptosis [312-315].